# Equilibrium shift, poisoning prevention, and selectivity enhancement in catalysis via dehydration of polymeric membranes

Myeong-Hun Hyeon[1,2,5], Hae-Gu Park[1,5], Jongmyeong Lee[1,5], Chang-In Kong[1], Eun-Young Kim[1,2], Jong Hak Kim [2], Su-Young Moon[1] ✉ & Seok Ki Kim [3,4] ✉

Generation of water as a byproduct in chemical reactions is often detrimental because it lowers the yield of the target product. Although several water removal methods, using absorbents, inorganic membranes, and additional dehydration reactions, have been proposed, there is an increasing demand for a stable and simple system that can selectively remove water over a wide range of reaction temperatures. Herein we report a thermally rearranged polybenzoxazole hollow fiber membrane with good water permselectivity and stability at reaction temperatures of up to 400 °C. Common reaction engineering challenges, such as those due to equilibrium limits, catalyst deactivation, and water-based side reactions, have been addressed using this membrane in a reactor.

Selective removal of byproducts from a chemical reaction can increase the yield of desired products by shifting the thermodynamic equilibrium toward the product side, according to the Le Chatelier's principle[1]. In addition, when a byproduct reacts with catalysts or reactants, removal of the byproduct can prevent catalyst poisoning or its side reactions with the reactant, ensuring long-term operation and enhanced product yields[2].

$H_2O$ is one of the major byproducts of various industrially mature and environment-friendly reactions, such as hydrocarbon combustion and $CO_2$- or biomass-derived chemical synthesis. Considerable efforts have been devoted toward the development of strategies for the selective removal of $H_2O$ during these reactions, including the insertion of a sorbent into a reactor[3] and use of an additional reactant that consumes $H_2O$[4]. The use of $H_2O$-permeable membranes has also been widely considered[5] because it does not require an additional process to regenerate sorbents or chemicals, allowing the design of a straightforward and energy-efficient chemical process.

Owing to the high thermal stability of inorganic materials like zeolites[6,7], silica[8], and metal oxides[9], these materials serve as membranes for selective $H_2O$ removal in various reactions. However, the $H_2O$ permselectivity of these membranes decreases significantly owing to the reduced pore-blocking effect of $H_2O$ at elevated temperatures[10]. The performances of the membranes reported previously are summarized in the Supplementary Information (Supplementary Fig. 1). The low thermal and chemical stability and the poor processability of various metal–organic framework (MOF)-based membranes developed for $H_2O$ separation limit their widespread application in membrane reactors[11]. However, a modified polyimide membrane has been recently reported to have moderate durability at the maximum reaction temperature of 300 °C, while retaining its $H_2O$ permselectivity, because of its high $H_2O$ solubility and diffusibility[12]. In addition, the facile processability and modulation characteristics of the polymeric material allowed it to be processed into hollow fiber membranes with a high surface-area-to-volume ratio, achieving a high permeation rate per unit volume of the reactor.

In this study, we developed a thermally rearranged polybenzoxazole (TR-PBO) membrane that was highly selective for $H_2O$ permeation up to a working temperature of 440 °C (Fig. 1). The corresponding membrane reactor was effective in shifting the thermodynamic equilibrium, preventing catalytic poisoning, and inhibiting

[1]C1 Gas & Carbon Convergent Research Center, Korea Research Institute of Chemical Technology, Daejeon 34114, Korea. [2]Department of Chemical and Biomolecular Engineering, Yonsei University, Seoul 03722, Korea. [3]Department of Chemical Engineering, Ajou University, Suwon 16499, Korea. [4]Department of Energy Systems Research, Ajou University, Suwon 16499, Korea. [5]These authors contributed equally: Myeong-Hun Hyeon, Hae-Gu Park, Jongmyeong Lee. ✉e-mail: msy1609@krict.re.kr; seokki@ajou.ac.kr

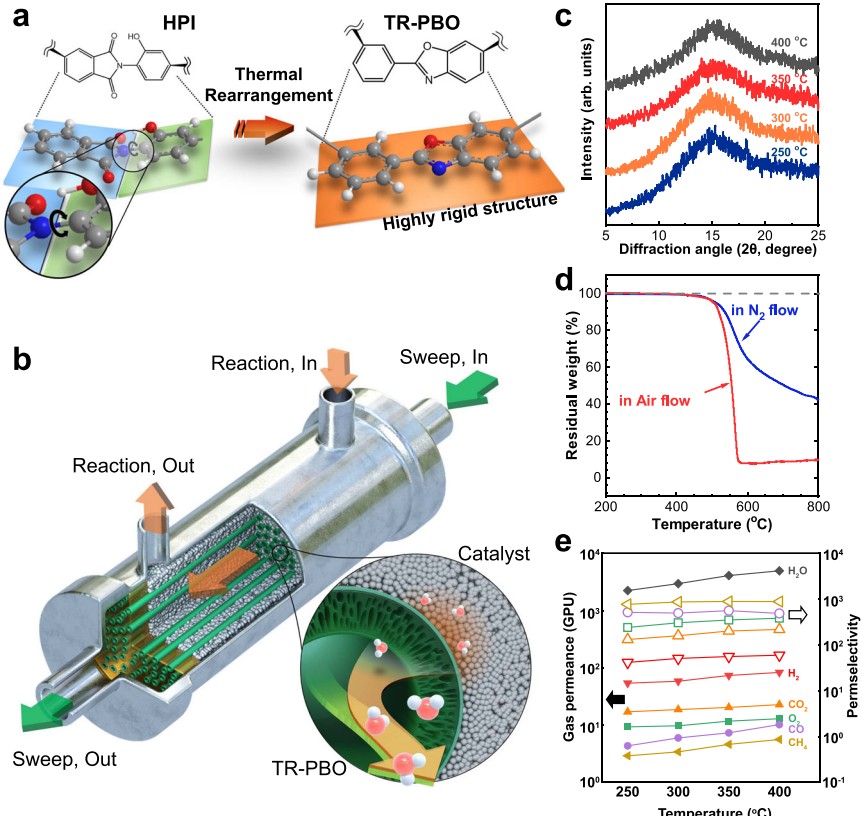

**Fig. 1 | Thermally rearranged polybenzoxazole (TR-PBO) membrane reactor.**
**a** Synthesis of TR-PBO from HPI. A highly rigid polymeric structure was obtained by the thermal rearrangement of HPI at 425 °C for 2 h under a N₂ atmosphere.
**b** Illustration of the membrane reactor module. TR-PBO hollow fibers were combined with a packed-bed catalyst reactor. H₂O molecules produced in the catalyst part selectively permeated into the bore side, where the sweep gas flows. **c** XRD pattern of TR-PBO at different temperatures. **d** TGA thermograms of TR-PBO under N₂ and air flows. **e** Permeability (in gas permeation unit, GPU) of $H_2O$, $H_2$, $CO_2$, $O_2$, CO, and $CH_4$ (solid symbols) and permselectivity of $H_2O$/gas (hollow symbols) through the TR-PBO membrane at different temperatures.

side reactions, as demonstrated using the reverse water–gas shift (RWGS) reaction, methane combustion reaction, and Fischer–Tropsch olefin (FTO) synthesis, respectively.

## Results and discussion
### Fabrication of TR-PBO membrane
The TR-PBO membrane was prepared in two steps: fabrication of hydroxyl polyimide (HPI) hollow fibers and simple heat treatment of HPI at 425 °C in N₂[13]. The synthetic procedure for HPI and the preparation method of the TR-PBO membrane are described in the Supplementary Information (Supplementary Methods and Supplementary Fig. 2). The HPI hollow fibers were fabricated by the dry-jet/wet-quench method using a home-built spinning setup (Supplementary Fig. 3 and Supplementary Table 1). The heat-treatment temperature was determined from the TGA profile of HPI (Supplementary Fig. 4 and Supplementary Method). This process has two main advantages. First, during the heat treatment, TR-PBO attained its unique porosity owing to the molecular rearrangement of HPI accompanying $CO_2$ evolution (Fig. 1a and Supplementary Fig. 2). The bimodal microcavity distribution of TR-PBO ($d_{small} \approx 0.3$ nm and $d_{large} \approx 0.8$ nm) enhanced the selectivity and permeability by the virtue of the size-exclusion effect and facile diffusion, respectively, which are not observed for conventional PBO. Second, even rigid and insoluble TR-PBO can be used as a material to form elaborate structures like hollow fibers (Fig. 1b and Supplementary Fig. 5) because the HPI precursors are easily processable, and thermal rearrangement does not destroy the membrane structure (Supplementary Fig. 6).

### Structural characterization of TR-PBO membrane
The thermal rearrangement of HPI to PBO was investigated through attenuated total reflectance infrared (ATR-IR) spectroscopy. The peaks corresponding to hydroxyl (O–H) stretching vibrations of the HPI precursors at ~2971 cm⁻¹ disappeared, while those corresponding to C=N stretching vibrations of the benzoxazole group appeared simultaneously at ~1063 cm⁻¹ in the spectrum of TR-PBO (Supplementary Fig. 7). The in situ X-ray diffraction (XRD) patterns of the TR-PBO hollow fibers at various temperatures are shown in Fig. 1c. The peak corresponding to the interchain distance in TR-PBO (2θ ≈ 15°) did not change till 400 °C, suggesting a rigid and thermally stable polymeric structure. This is in accordance with the differential scanning calorimetry thermogram (Supplementary Fig. 8), which did not show any glass transition temperature ($T_g$) for TR-PBO up to 400 °C. Thermogravimetric analysis (TGA, Fig. 1d) indicated that TR-PBO did not degrade up to 500 °C in either N₂ or air atmosphere. The isotherm at 400 °C also indicated the high thermal stability of TR-PBO (Supplementary Fig. 9). Given that the maximum available operation temperature of HPI is 300 °C[12], TR-PBO is more suitable for high-temperature applications. The ultrahigh thermal stability of TR-PBO is attributed to its rigid rod-like structure, which has a high torsional energy barrier for rotation between the heterocyclic phenylene rings (Fig. 1a).

### Transport behavior through the TR-PBO membrane
The transport behaviors of the reactant and product gases involved in the present study were evaluated at 250–400 °C (Fig. 1e and Supplementary Fig. 10). The permeance of each gas increased with the

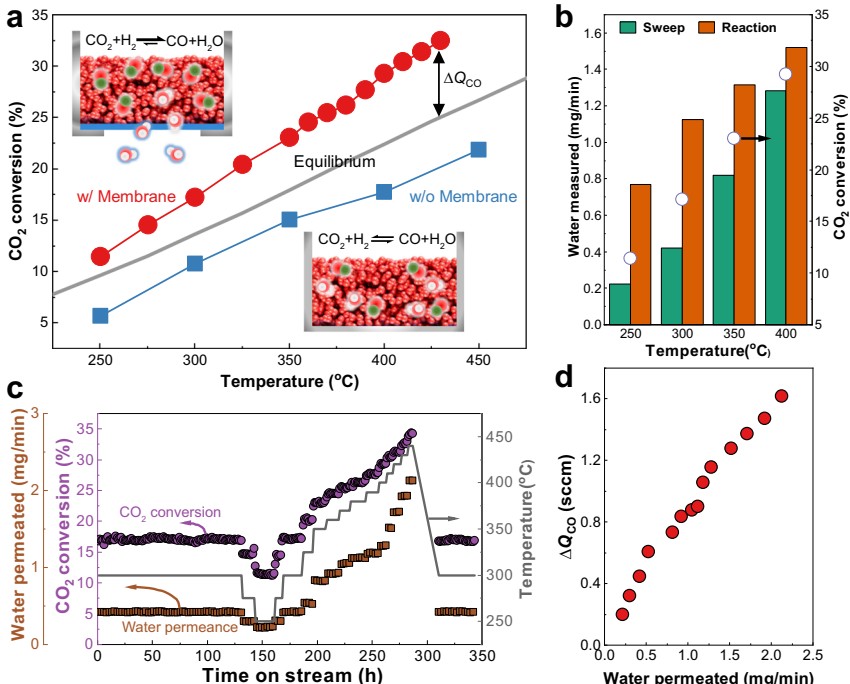

**Fig. 2 | Equilibrium-shift effect in the RWGS reaction. a** $CO_2$ conversion as a function of temperature with (w/) and without (w/o) the membrane. **b** $CO_2$ conversion and amount of $H_2O$ measured at the outlet of the sweep and reaction parts. **c** $CO_2$ conversion and amount of $H_2O$ permeated from the reaction part to sweep part as a function of time on stream. **d** Additional amount of CO produced beyond equilibrium ($\Delta Q_{CO}$) as a function of the amount of $H_2O$ permeated from the reaction part to sweep part. Reactions were carried out using 6.4 g $CuZn/Al_2O_3$ catalyst under atmospheric pressure. Reaction and sweep gases flowed at 24 sccm with an identical composition of $H_2/CO_2$ ratio of 1 (gas hourly space velocity = 225 mL/$g_{cat}$ h).

temperature. The extent of increase depended on the kinetic diameter of the penetrant (Supplementary Fig. 11); in other words, gas permeation through the TR-PBO membrane was governed by the diffusion of a gas molecule. The permselectivity of $H_2O$ to other gases tended to increase with temperature because the increase in the permeability of $H_2O$ was more distinct than those of other large molecules. In addition, the permeability of mixed gases was evaluated, as shown in Supplementary Fig. 12, which shows that $H_2O$/gas permselectivities of single and mixed gases were different to some extent. This could be attributed to the nonideal permeation dynamics of the mixed gas test, wherein the sorption and diffusion of gases occur competitively.

For inorganic membranes, such as zeolites, selective $H_2O$ permeation originates from their hydrophilicity, which facilitates pore blocking with condensed $H_2O$ and inhibits the penetration of other gases[10]. However, at elevated temperatures, the hydrophilicity decreases, and the pore-blocking effect diminishes to the extent that the selective $H_2O$ permeation disappears. As shown in Supplementary Fig. 1, among the polymeric and inorganic membranes reported to date, the TR-PBO hollow fibers exhibited the highest $H_2O$ permselectivity at 250–400 °C.

## Application of the TR-PBO membrane reactor
The performance of the TR-PBO membrane reactor for the RWGS reaction, methane combustion reaction, and FTO reaction was determined on home-built equipment (Supplementary Fig. 12 and Supplementary Methods for reactions). Each catalytic reaction was carried out under identical conditions, with or without the membrane.

## Shifting the thermodynamic equilibrium
The use of the fabricated membrane for the in situ separation of $H_2O$ could drive a reaction beyond its thermodynamic limit, as evident from the reversible and equilibrium-controlled RWGS reaction. In this

reaction, we used a commercial CuZn catalyst that exhibited a CO selectivity of over 99% during the reaction. Figure 2a shows that the use of the TR-PBO membrane (w/membrane) increased the $CO_2$ conversion beyond the thermodynamic equilibrium, while the membrane-free reaction (w/o membrane) resulted in a slightly lower $CO_2$ conversion than that predicted by the equilibrium curve. As expected from the gas permeance test (Fig. 1e), the amount of $H_2O$ permeated from the catalyst bed to the sweep side through the hollow fiber increased with increasing reaction temperature (Fig. 2b). The $CO_2$ conversion and $H_2O$ permeation as a function of time on stream are shown in Fig. 2c. As the reaction temperature increased from 250 to 440 °C in approximately 350 h, the TR-PBO membrane exhibited stable performance that was maintained regardless of the temperature history. The additional amount of CO produced beyond equilibrium ($\Delta Q_{CO}$) increased proportionally with increasing amount of $H_2O$ permeated (Fig. 2d), indicating that the increased $CO_2$ conversion resulted from the $H_2O$ removal.

The performance of a membrane reactor varies depending on the composition or flow rate of the sweep gas[12,14]. Figure 2 shows the results of the reaction conducted using a sweep gas whose composition and flow rate were identical to those of the reaction gas. The results obtained using other sweep gas compositions are shown in Supplementary Fig. 14. In the absence of the sweep gas, the $CO_2$ conversion was similar to that in the case without the membrane. When pure $N_2$ was used as the sweep gas, the $CO_2$ conversion was lower than that obtained without the membrane. However, we believe that this is not a problem with the membrane itself, but rather a challenge from a reactor design perspective. Supplementary Fig. 12 shows that in the presence of $H_2O$, the permeabilities of $CO_2$ and $H_2$ decrease by an order of magnitude compared to their pure-gas permeabilities, while $H_2O/H_2$ and $H_2O/CO_2$ permselectivities are maintained. This suggests that $CO_2$ and $H_2$ crossover occurs in the upper part of the reactor before the reaction gas meets the catalyst. This phenomenon is likely

more apparent in the lab-scale reactor used in this study due to the small volume of the catalyst bed, as shown in Supplementary Fig. 13. However, this issue is expected to be minimized in large-scale reactors, where the catalyst bed occupies the majority of the reactor volume, leaving only a small dead space above the catalyst bed where only membranes are present. Moreover, this issue can be mitigated by using a sweep gas with the same composition as the reactants, reducing the difference in reactant concentrations on both sides of the membrane and preventing the reactants from escaping while selectively removing generated $H_2O$ molecules. In this case, a simple gas-liquid separator can be used in practical applications to continuously separate water from the gas and recycle the sweep gas. In Supplementary Table 2, reactor volume of the membrane reactor is compared with that of packed-bed reactor for a commercial scale production of CO. The membrane reactor required less volume than the packed-bed reactor at the same feed flow rate.

## Prevention of catalytic poisoning

Pd-based catalysts are effective for low-temperature methane combustion[15]. However, $H_2O$ generated during the reaction readily binds to the active sites of the catalysts and significantly reduces the combustion performance[3]. We demonstrated a TR-PBO membrane reactor to prevent $H_2O$-induced deactivation of a Pd catalyst during methane combustion.

We used an $Al_2O_3$-supported Pd catalyst that exhibited an initial $CH_4$ conversion of ~95% without the use of a membrane reactor (Fig. 3a). As the reaction progressed, the Pd catalyst lost its initial activity, and the $CH_4$ conversion decreased linearly to ~80% after 150 h on stream. Within the same period, the reaction conducted using the TR-PBO membrane exhibited almost constant activity, maintaining the initial $CH_4$ conversion of ~96% even after 150 h.

Figure 3b shows the amount of water measured at the outlet of the membrane reactor. Approximately 40% of the water generated in the reaction was removed by the TR-PBO membrane. The presence of

water deteriorates the catalytic performance by forming Pd hydroxyls and eventually delays the regeneration of PdO, which serves as the active phase during methane oxidation[16]. We analyzed the Pd 3d chemical states of the spent catalysts using X-ray photoelectron spectroscopy (XPS) and compared the profiles with that of the fresh catalyst (Fig. 3c). The catalyst used in the membrane reactor showed Pd oxide peaks that were almost identical to those of the fresh catalyst. In contrast, the catalyst used without the membrane showed peaks corresponding to metallic Pd, indicating the irreversible phase transition of PdO using nondissipated $H_2O$. It is noteworthy that the use of $N_2$ as the sweeping gas did not show a negative effect as was the case with RWGS (Supplementary Fig. 14). This is attributed to the impermeability of the reactants, owing to which they completely pass through the catalyst layer without any crossover, despite the significant difference in the reactant concentrations on both the sides of the membrane.

## Suppression of side reactions

Finally, a Fischer−Tropsch reaction to produce olefins was carried out in the TR-PBO membrane reactor using a Fe catalyst to demonstrate the suppression of side reactions upon $H_2O$ removal. As shown in Fig. 4a, the oxygen atom of CO is removed as $H_2O$, which reacts with CO again [the well-known water–gas shift (WGS) reaction]. The accompanying WGS reaction significantly lowered the olefin yield and produced $CO_2$. The amount of CO consumed to produce $CO_2$ during typical FTO synthesis processes is reportedly >40%[17,18].

The effect of using the TR-PBO membrane to the Fischer−Tropsch reaction is shown in Fig. 4b−d. Although the Fischer−Tropsch reaction tested in this study was not limited by the thermodynamic equilibrium, the amount of CO converted over the Fe catalyst increased (Fig. 4b) upon using the TR-PBO membrane that removed the produced $H_2O$ (Fig. 4c). In situ water removal improved the hydrocarbon selectivity significantly (Fig. 4d); in other words, the amount of $CO_2$ produced by the WGS reaction reached 47.9% in the absence of the membrane but decreased to 15.1% when the TR-PBO membrane was used. This

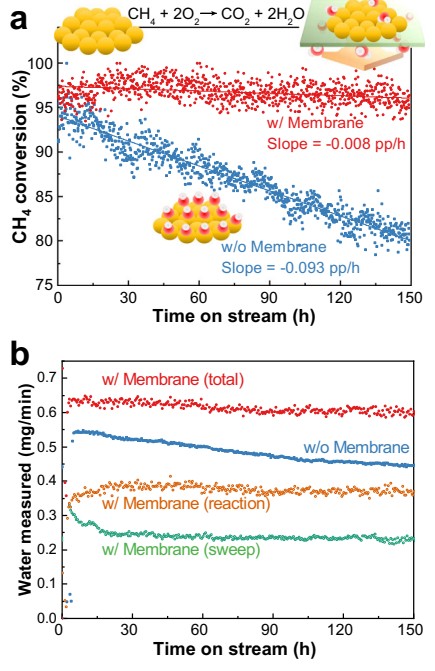

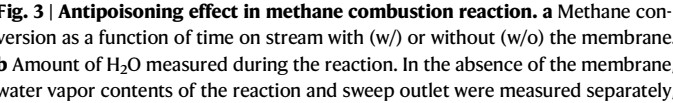

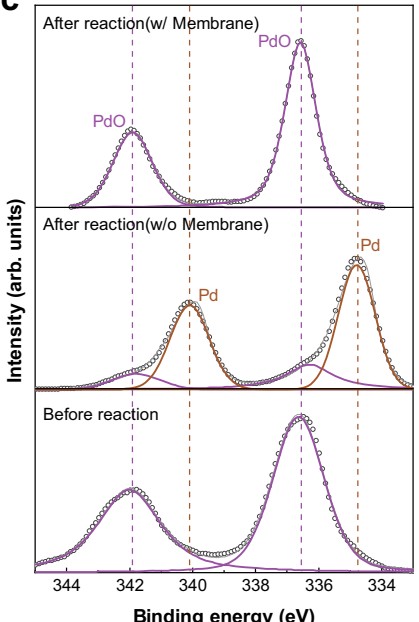

**Fig. 3 | Antipoisoning effect in methane combustion reaction. a** Methane conversion as a function of time on stream with (w/) or without (w/o) the membrane. **b** Amount of $H_2O$ measured during the reaction. In the absence of the membrane, water vapor contents of the reaction and sweep outlet were measured separately, and the sum was denoted as total. **c** Pd 3d XP spectra of the catalysts before and after the reaction. Reactions were carried out using 1 g $Pd/Al_2O_3$ at $T = 300\,°C$ and $P = 0.1\,MPa$. The reaction gas was flowed at 83.3 sccm and comprised 0.4% $CH_4$ and 4% $O_2$ in Ar balance. The sweep gas was pure $N_2$ and was flowed at 83.3 sccm.

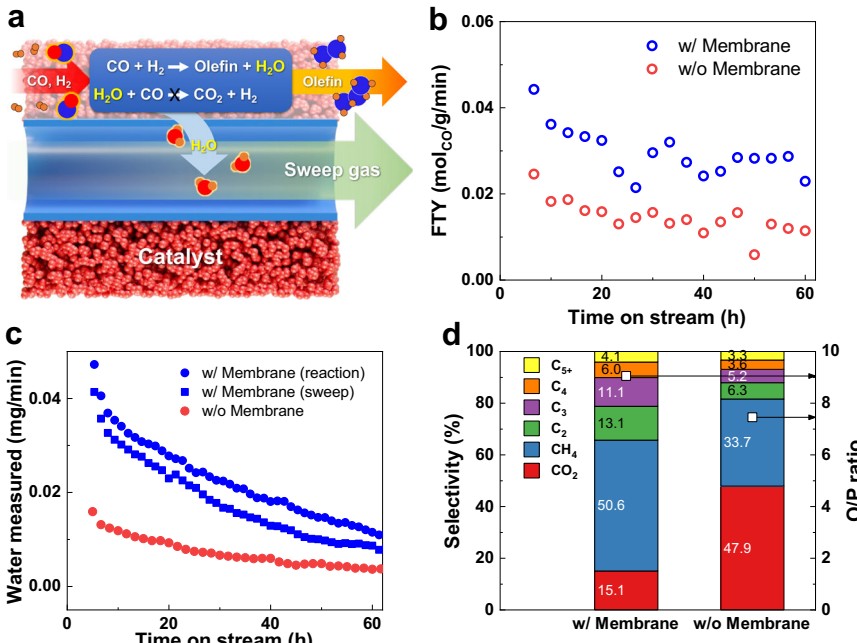

**Fig. 4 | Suppression of side reactions in Fischer–Tropsch (FT) synthesis.**
**a** Suppression of the water–gas shift reaction during FT synthesis using the TR-PBO membrane reactor. **b** Fe time yield (FTY) for the Fischer–Tropsch reaction conducted with (w/) and without (w/o) the TR-PBO membrane. **c** Amount of $H_2O$ measured during the reaction. In case of w/o membrane, moisture in the reaction and sweep outlet was measured separately. **d** Product selectivity measured at a time on stream of 40 h. Olefin-to-paraffin ratio (O/P ratio) of $C_2$–$C_4$ hydrocarbons are shown as squares. Reactions were carried out using 1 g KFeCuAl catalyst at $T = 320\,°C$ and $P = 0.1\,MPa$. The reaction gas flowed at 30 sccm and comprised 30% CO and 60% $H_2$ in Ar balance. The sweep gas was fed at composition and flow rate identical to those of the reaction gas.

indicates that the absence of $H_2O$ in the catalyst bed prevents further WGS reactions.

In addition to the selective conversion of CO to hydrocarbons, suppression of the WGS reaction increases the olefin selectivity. Figure 4d shows that the olefin-to-paraffin ratio of light hydrocarbons ($C_2$–$C_4$) was higher with the membrane than without it. This is because the production of hydrogen, which saturates olefins, decreases with the suppression of the WGS reaction.

The TR-PBO membrane reactor developed herein facilitated effective $H_2O$ removal in various reactions where $H_2O$ inhibits the catalytic performance. Through the equilibrium shift and the suppression of deactivation and side reactions, the target-product yields improved significantly, which typically may not be achievable with catalyst developments. The as-developed membrane reactor exhibited thermal, chemical, and mechanical stability for a long time on stream in various high-temperature reactions, thereby elevating the industrial prospects of such polymeric membrane reactors. Future studies can be conducted on investigating the effects of the membrane area, flow configuration, flow rate, and recirculation process on the separation-performance metrics, with a view to optimizing and upscaling the membrane reactor and process.

## Methods
### Fabrication of TR-PBO hollow fibers and their modulation
TR-PBO fibers were prepared by heat treatment of HPI precursor fibers, as shown in the schematic in Supplementary Fig. 2b. The synthesis of HPI hollow fibers is described in Supplementary Methods. Each HPI fiber was placed in 70-cm-long honeycomb-shaped ceramic tubes and treated at 300 °C for 1 h in a cylindrical furnace to remove residual solvents and fully convert the unreacted amic acid into imide groups. Then, the HPI fibers were heated to 425 °C at a ramping rate of 5 °C/min and dwelled for 30 min, followed by cooling to the ambient temperature. The diameter and wall thickness of the TR-PBO fibers were measured to be 487.2 and 48.4 μm, respectively (Supplementary Fig. 6b).

For hollow fiber modulation, a membrane module housing was assembled using ½-inch-diameter stainless steel pipes. As shown in Supplementary Fig. 5, both ends of a 34-cm-long pipe were connected to two three-way stainless steel connectors, and two 7-cm-long pipes were connected to both ends of the connectors. A bundle of 40 TR-PBO fibers was inserted into the housing, and the tips of the fibers were presealed to prevent the external epoxy resin from intruding into the hollow fibers. The epoxy resin was filled in the two short pipes and hardened for 12 h. Finally, the hardened epoxy resins stuck outside the stainless steel pipes were cut. The effective area of the prepared TR-PBO module was 210 cm².

### Reaction 1: Reverse water–gas shift
For the RWGS reaction, the CuZn/Al$_2$O$_3$ catalyst was chosen owing to the high $CO_2$ conversion and >99.9% CO selectivity. First, to confirm the equilibrium activity of the neat catalyst, we conducted a RWGS reaction using a fixed-bed stainless steel reactor. Zirconia balls (diameter = 0.3 mm) used to support the catalyst layer were prepacked in an empty reactor. Next, 6.4 g of the catalyst powder of 30–40 mesh size (sieved by a mesh sieve of 420–560 mesh size) was stacked on the zirconia layer. The catalyst-containing reactor was installed vertically through a small cylindrical furnace in which the catalyst layer was positioned at the center of the furnace. Before the reaction, the reactor was heated to 300 °C for 3 h under a $H_2$ atmosphere for catalyst reduction. For the RWGS reaction, a feed $H_2/CO_2$ stream with a 1:1 molar ratio (12.0/12.0 sccm) was continuously supplied at a space velocity of 225 mL/g$_{cat}$·h under atmospheric pressure.

For the RWGS reaction in the membrane module, a zirconia ball and catalyst were filled into a TR-PBO hollow fiber module. The measurement procedure was identical to that described in Supplementary Methods, except that the $H_2/CO_2$ stream was used as the sweep gas (see Supplementary Fig. 12b). The feed (reaction)/sweep flow rate ratio was fixed at 1:1. $N_2$ gas was used as an

internal standard gas for gas chromatography (GC) analysis and injected at a flow rate of 12.0 sccm just behind the sweep-out and feed-out sides.

### Reaction 2: Methane combustion

The equilibrium activity of the neat catalyst for methane combustion was measured using a palladium oxide (PdO) catalyst. Unlike Reaction 1, SiC powder was used to support the PdO catalyst bed (1.0 g); before the reaction, the catalyst was reduced at 300 °C for 1 h under a $H_2$ atmosphere and additionally oxidized at 300 °C for 1 h under air purging. For methane combustion, the feed gas composed of $O_2$ (4%), $CH_4$ (0.4%), and Ar (balanced) was fed at 83.3 sccm to the feed-in side at a space velocity of 5000 mL/$g_{cat}$·h. For GC analysis, He as an internal gas was fed at a flow rate of 10 sccm just behind the feed-out side. During the reaction for 150 h, the temperature was fixed at 300 °C, and the product gas at the feed-out side was characterized using a GC instrument and an electric mass flowmeter.

The methane combustion reaction was performed using the same procedure as above, using a catalyst-filled TR-PBO fiber module with $N_2$-sweep at a flow rate of 83.3 sccm. He was fed as an internal gas at a flow rate of 10 sccm immediately behind both the feed-out and sweep-out sides.

### Reaction 3: Fischer–Tropsch olefin synthesis

$K_4Fe_{100}Cu_6Al_{16}$ was used as a catalyst for the FTO synthesis. The equilibrium activity of the neat catalyst was evaluated on the experimental setup illustrated in Supplementary Fig. 12a using 1 g of $K_4Fe_{100}Cu_6Al_{16}$ catalyst with SiC as the supporting layer material. Before the reaction, the catalyst was reduced at 320 °C for 1 h under a $H_2$ atmosphere and ambient pressure. A mixture of CO (30%), $H_2$ (60%), and Ar (10%) gases was introduced as the feed gas into the fixed-bed reactor at a space velocity of 1800 mL/$g_{cat}$·h (30 sccm). The Fischer–Tropsch reaction was conducted at 320 °C, and the product gases were analyzed using GC. For the GC analysis, gaseous products (CO, $CO_2$, and $CH_4$) and hydrocarbons (C1–C4) were analyzed using a thermal conductivity detector (TCD) and flame ionization detector (FID), respectively. $N_2$ (10 sccm) was used as the internal gas for GC analysis.

A Fischer–Tropsch reaction was conducted in a catalyst-filled TR-PBO membrane module on the experimental setup shown in Supplementary Fig. 12b using the same procedure as described above. The same composition as that of the feed gas was fed as a sweep gas at a flow rate of 30 sccm. $N_2$ was fed as an internal gas at a flow rate of 10 sccm just behind the feed-out and sweep-out sides. The gas components on both sides were analyzed using GC.

### Data availability

All data are available in the main text or the supplementary information.

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

### Acknowledgements

This work was supported by the Korea Research Institute of Chemical Technology (KRICT) [grant no. BSF22-504], Korea Institute of Energy Technology Evaluation and Planning and the Ministry of Trade, Industry & Energy of the Republic of Korea [grant no. 20224C10300010], and GRRC program of Gyeonggi province [grant no. GRRC AJOU 2022-B02].

### Author contributions

S.-Y.M. and S.K.K. conceived the project and jointly supervised the study. J.L. synthesized membranes. H.-G.P. synthesized catalysts. M.-H.H. performed catalyst reaction analyze data. J.L., H.-G.P., M.-H.H., S.-Y.M., and S.K.K. designed the membrane reactor. C.-I.K. and E.-Y.K. helped with characterizations. J.H.K. discussed the results and commented on the original draft. J.L., H-G.P., and M.-H.H wrote the original manuscript. S.-Y.M. and S.K.K. revised the manuscript. All authors commented on the manuscript.

### Competing interests

The authors declare no competing interests.
