## [Peer Review File · Nature Communications]

Equilibrium shift, poisoning prevention, and selectivity enhancement in catalysis via dehydration of polymeric membranesREVIEWER COMMENTS

Reviewer #1 (Remarks to the Author):

This is a nice manuscript that reports the synthesis of a thermally rearranged polybenzoxazole (TR-PBO) membrane with high selectivity for H₂O permeation up to a working temperature of 440 °C. The manuscript provides the structural characterization of the membrane and the transport performance is analysed with three well-known and high interest reactions, i.e., 1) the reversible water gas shift reaction where the membrane reactor, provides conversion beyond equilibrium, 2) methane combustion to prevent catalyst poisoning and, 3) Fisher-Tropsch olefin formation to show the influence of suppression of side reactions.

Next, some comments are expressed,

1. The title of the manuscript is too general in a topic that has been widely referenced; it is difficult to guess what is the new contribution. Could it be more specific?
2. Figure 1E provides the data of membrane selectivity to different gas molecules obtained from permeation data of gas streams with individual components. These data should be corroborated working with mixed gas samples. The values of selectivity can not be read in the y axis.
3. The influence of the sweep gas, as shown in Fig S13, is troublesome. The only way to keep the performance of the membrane reactor is by feeding a sweep gas with the same composition as the feed gas, what is the loss of reactants operating in this way, how is water removed from the sweep gas?. If no sweep gas is used the performance of the MR remains similar to the reactor without membrane, so there is no point to complicate the reactor design.
4. Why there is no evidence of the negative effect of sweep gas (nitrogen at 83.3 sccm) in Fig 3B as it was observed in Fig S13?

Reviewer #2 (Remarks to the Author):

This is an interesting paper with a water-permeable membrane being applied to multiple reaction engineering problems within the same piece of work. The reaction engineering concepts are not new as the authors acknowledge. However, the membrane may well be innovative (unfortunately this reviewer is not an expert on this class of membrane).

Unfortunately, the level of rigour used in the analysis of the reaction case studies is well below the standard required. Taking the study of the RWGS reaction, superficially interesting results are shown with CO₂ conversions apparently increased above the equilibrium line when the membrane is present. However, there is a concern about how robust these results are. Firstly the equilibrium line should be reproduced experimentally by varying residence time. This confirms that operation is as expected (could be performed both with and without a membrane present). However it is also very important to

compare the water flux through the membrane with the excess carbon dioxide conversion to assure the reader that operation is as expected. C and H mole balances should also be undertaken. It is important to note that if one checks the SI in addition to the paper, the SI does not address any of these points.

The same level of rigour should be applied to the other reaction examples (but the experiments required will vary due to the different nature of the reaction mechanisms being explored).

Reviewer #3 (Remarks to the Author):

The manuscript introduces a new membrane formulation (TR-PBO) for in-situ removal of water with exceptional thermal stability, H₂O permeability, and H₂O perm-selectivity. The results are novel and interesting and should be published after the following comments are addressed:

Major comments:

1- Report the water permeance of membrane in mol/m²/s/Pa or Barrer with membrane thickness instead of GPU to be able to easily compare the performance of the membrane to those reported in the literature.

2- For the RWGS reaction, add a figure to plot CO₂ conversion as a function of temperature (H₂O permeance of the membrane) and weight hourly space velocity (WHSV) of the membrane reactors.

3- Fig S.13: It does not make sense that when the sweep gas is changed to N₂, the CO₂ conversion drops significant below thermodynamic limits. As shown Table S2, the H₂O perm-selectivity of the membrane with respect to H₂ and CO₂ is close to 400, and 1400, respectively. These perm-selectivities do not seem to result in escaping of the reactants (H₂ and CO₂) to the sweep to cause lower CO₂ conversion. Authors need to clearly justify the experimental results of N₂ sweep. It might not be practical to use the sweep gas with the same composition of the feed, especially when the sweep gas involves the expensive H₂ in the RWGS reaction.

4- What is the effect of counter-current sweep to feed configuration on the CO₂ conversion of the RWGS membrane?

5- Since membrane processes are slow, they need larger volumes to produce products. Authors need to compare the volume of a RWGS membrane reactor to that of a conventional packed-bed RWGS reactor to produce 1000 kmol/hr of syngas to be able to judge the practicality of membrane reactors at large scales in terms of size (umber of tubes, tube diameter, etc) and cost.

Minor comments:

1- Add a nomenclature to the manuscript.

2- Fig S1: Instead of references in brackets, show the actual name of the membranes on the figures.

3- Fig. 1: The numbers of the y-axis on the right-hand side of Fig. 1E should be added for the H₂O-permselectivity. There is no number on the right hand side y-axis.

4- Some figures are not labeled properly. For example, in Fig. 2 C, two graphs are pointed to the left y-axis. Which set of data is for CO₂ conversion and which set of data is for H₂O permeance?

Thank you for this opportunity to improve our manuscript; we have addressed the reviewers' comments in this letter and in the attached document. The reviewers' comments (verbatim) and our point-by-point responses have been indicated in blue and black, respectively. The revised text has been indicated in red.

Reviewer #1 (Remarks to the Author):

Comments:

This is a nice manuscript that reports the synthesis of a thermally rearranged polybenzoxazole (TR-PBO) membrane with high selectivity for H₂O permeation up to a working temperature of 440 °C. The manuscript provides the structural characterization of the membrane and the transport performance is analyzed with three well-known and high interest reactions, i.e., 1) the reversible water gas shift reaction where the membrane reactor, provides conversion beyond equilibrium, 2) methane combustion to prevent catalyst poisoning and, 3) Fisher-Tropsch olefin formation to show the influence of suppression of side reactions.

Next, some comments are expressed,

1. The title of the manuscript is too general in a topic that has been widely referenced; it is difficult to guess what is the new contribution. Could it be more specific?

As per your comment, we have changed the title from "*In situ water removal in a polymeric membrane reactor during catalysis*" to "*Equilibrium shift, poisoning prevention, and selectivity enhancement in catalysis via dehydration of polymeric membranes*" to render it more specific and representative of the study.

Title:

Equilibrium shift, poisoning prevention, and selectivity enhancement in catalysis via dehydration of polymeric membranes

2. Figure 1E provides the data of membrane selectivity to different gas molecules obtained from permeation data of gas streams with individual components. These data should be corroborated working with mixed gas samples. The values of selectivity can not be read in the y axis.

Thank you for pointing out an important aspect related to the gas permeation performance of the membrane. To precisely distinguish the permeation performance of each gas, the single-gas permeability has been shown in Fig. 1e, whereas the mixed-gas permeability has been added to Supplementary Fig. 12 in Supplementary Information. As inferred from Supplementary Fig. 12, there are some differences between the single- and mixed-gas permeabilities, but high H₂O permselectivities toward other gases are still maintained. The following description has been added to the text:

(In the section "*Transport behavior through the TR-PBO membrane*")

The permselectivity of H₂O to other gases tended to increase with temperature because the increase in the permeability of H₂O was more distinct than those of other large molecules. Additionally, the permeability of mixed gases was evaluated, as shown in Supplementary Fig. 12, which shows that H₂O/gas permselectivities of single and mixed gases were different to some extent. This could be attributed to the nonideal permeation dynamics of the mixed gas test, wherein the sorption and diffusion of gases occur competitively.

(In Supplementary Information)

Supplementary Fig. 12. Comparison of permeabilities of **a** single gas and **b** mixed gas. The solid and hollow symbols denote the permeance and H₂O/gas permselectivity, respectively. The mixed gas permeation test was performed using RWGS reaction setup, where sweep gas was not fed.

Additionally, we have modified Fig. 1e by specifying values for the right-hand y-axis for clarity.

Fig. 1

3. The influence of the sweep gas, as shown in Fig S13, is troublesome. The only way to keep the performance of the membrane reactor is by feeding a sweep gas with the same composition as the feed gas, what is the loss of reactants operating in this way, how is water removed from the sweep gas? If no sweep gas is used the performance of the MR remains similar to the reactor without membrane, so there is no point to complicate the reactor design.

We agree with your concern; we have taken a lot of trial and error to find a solution that uses the sweep gas with the same composition as the reaction gas enabling the membrane reactor to overcome the thermodynamic limit in this reaction. Moreover, the performance of the membrane reactor depends on the type of the sweep gas. We selected the RWGS as a representative reaction to demonstrate the capability of the membrane in producing the equilibrium-shift effect, and we considered that the practical limitation is worth mentioning. When the RWGS reaction was performed with an inert sweeping gas, the permeabilities of the feed gases (H₂ and CO₂) were sufficiently high; thus, the reactants considerably penetrated the membrane before reacting on the catalyst bed and reaching the sweeping part. The N₂ sweeping gas boosted the permeance of H₂ and CO₂ by preventing the concentration polarization on the membrane surface. In the

case of N₂ sweeping, the crossover amounts of H₂ and CO₂ were 49%–62% and 46%–54%, respectively. To minimize this escaping phenomenon, we used a sweep gas with the same composition as the reaction gas. Consequently, the concentration difference between the two sides of the membrane was reduced, and gas permeation was prevented. Currently, the process is being optimized via adjustment of the catalyst position and fabrication of locally impermeable membranes. For practical applications, the process efficiency can be increased by trapping the moisture and recycling the sweeping gas. The explanation has been added to the text as follows:

(In the section “*Shifting the thermodynamic equilibrium*”)

When pure N₂ was supplied as the sweeping gas, the CO₂ conversion was lower than that obtained without the membrane because the N₂ sweeping accelerated the partial permeation of the small reactant molecules through the membrane, before reacting on the catalyst bed. By using a sweeping gas with the same composition as the reactants, the difference between the reactant concentrations on both the sides of the membrane could be reduced, thereby preventing the reactants from escaping and selectively removing the generated H₂O molecules. In practical applications, a simple moisture trap can be used to recycle the sweeping gas.

4. Why there is no evidence of the negative effect of sweep gas (nitrogen at 83.3 sccm) in Fig 3B as it was observed in Fig S13?

Unlike RWGS, the use of nitrogen in the CH₄ combustion did not result in any negative effect, because the reactant molecules (CH₄ and O₂) are sufficiently large to not permeate the membrane. This description has been added to the relevant text, as follows:

(In the section “*Prevention of catalytic poisoning*”)

It is noteworthy that the use of N₂ as the sweeping gas did not show a negative effect as was the case with RWGS (Supplementary Fig. 14). This is attributed to the impermeability of the reactants, owing to which they completely pass through the catalyst layer without any crossover, despite the significant difference in the reactant concentrations on both the sides of the membrane.

Reviewer #2 (Remarks to the Author):

Comments:

This is an interesting paper with a water-permeable membrane being applied to multiple reaction engineering problems within the same piece of work. The reaction engineering concepts are not new as the authors acknowledge. However, the membrane may well be innovative (unfortunately this reviewer is not an expert on this class of membrane).

Thank you for your positive comments.

Unfortunately, the level of rigour used in the analysis of the reaction case studies is well below the standard required. Taking the study of the RWGS reaction, superficially interesting results are shown with CO₂ conversions apparently increased above the equilibrium line when the membrane is present. However, there is a concern about how robust these results are. Firstly the equilibrium line should be reproduced experimentally by varying residence time. This confirms that operation is as expected (could be performed both with and without a membrane present). However it is also very important to compare the water flux through the membrane with the excess carbon dioxide conversion to assure the reader that operation is as expected. C and H mole balances should also be undertaken. It is important to note that if one checks the SI in addition to the paper, the SI does not address any of these points.

The same level of rigour should be applied to the other reaction examples (but the experiments required will vary due to the different nature of the reaction mechanisms being explored).

We agree with your concern. Accordingly, we performed additional experiments to confirm that the RWGS reaction proceeds under equilibrium-limited conditions. We conducted the reaction without the membrane at three different space velocities of the feed gas, which were higher than that applied during the membrane-assisted reaction; the CO₂ conversion was the same under all three conditions. Hence, the reaction occurred under equilibrium-limited conditions in the membrane reactor, and the increase in the conversion is attributed to the H₂O removal. The test results have been added to SI as follows:

(In Supplementary Information)

Supplementary Fig. 15. CO₂ conversion in the RWGS reaction conducted using a fixed-bed reactor without the membrane. The reactions were tested at three different space velocities: 1000, 2000, and 6000 mL/g_{cat}·h. The reactor has dimensions identical to that of the membrane reactor, including the catalyst bed configuration. The CO₂ conversion was almost the same under all three conditions, suggesting that all the reactions occurred close to the experimental equilibrium, which is 3–5% lower than the theoretical equilibrium conversion (blank circles). Because a lower space velocity was applied in the membrane reactor, the increase in the CO₂ conversion in the membrane reactor is attributed to the equilibrium shift due to H₂O removal.

Since the methane combustion and the Fischer–Tropsch reaction are not limited by equilibrium and have not been conducted with the aim of overcoming the equilibrium limit, the tendencies of these reactions to approach the near-equilibrium region were not verified.

Following your suggestion, we have added the carbon balance along with the calculation method in SI as follows:

(In Supplementary Information, Note 4. Reaction 1: Reverse Water–Gas Shift)

The carbon balance of the RWGS shift was calculated as follows:

$$C \text{ balance}(\%) = \frac{\text{Amounts of } CO_2 \text{ and } CO \text{ detected out of the reactor}}{\text{Amount of } CO_2 \text{ fed into the reactor}} \times 100 \quad \text{equation (7).}$$

(In Supplementary Information, Note 5. Reaction 2: Methane Combustion)

The carbon balance of the CH₄ oxidation was calculated as follows:

$$C \text{ balance}(\%) = \frac{\text{Amounts of } CH_4 \text{ and } CO_2 \text{ detected out of the reactor}}{\text{Amount of } CH_4 \text{ fed into the reactor}} \times 100 \quad \text{equation (11).}$$

(In Supplementary Information, Note 6. Reaction 3: Fischer–Tropsch Olefin Synthesis)

The selectivity of the Fischer–Tropsch reaction was calculated as

$$CO_2 \text{ selectivity}(\%) = \frac{\text{Amount of } CO_2 \text{ produced}}{\text{Amount of } CO \text{ converted}} \times 100 \quad \text{equation (14),}$$

$$C_{1-4} \text{ selectivity}(\%) = \frac{\text{Amount of } C_{1-4} \text{ hydrocarbons produced}}{\text{Amount of converted } CO} \times 100 \quad \text{equation (15),}$$

$$C_{5+} \text{ hydrocarbons selectivity}(\%) = 100 - C \text{ balance} \quad \text{equation (16).}$$

The carbon balance of the Fischer–Tropsch olefin synthesis was calculated using

$$C \text{ balance}(\%) = \frac{\text{Amounts of } CO, CO_2, C_{1-4} \text{ detected out of the reactor}}{\text{Amount of } CO \text{ fed into the reactor}} \times 100 \quad \text{equation (17).}$$

(In Supplementary Information)

Supplementary Fig. 16. Carbon balances of the **a** RWGS with the membrane, **b** RWGS without the membrane, **c** CH₄ oxidation with the membrane, **d** CH₄ oxidation without the membrane, **e** FTS with the membrane, and **f** FTS without the membrane. The carbon balance calculations for the RWGS, CH₄ oxidation, and FTS reactions are provided in equations (7), (11), and (17), respectively.

Reviewer #3 (Remarks to the Author):

Comments:

The manuscript introduces a new membrane formulation (TR-PBO) for in-situ removal of water with exceptional thermal stability, H₂O permeability, and H₂O perm-selectivity. The results are novel and interesting and should be published after the following comments are addressed:

Major comments:

1. Report the water permeance of membrane in mol/m²/s/Pa or Barrer with membrane thickness instead of GPU to be able to easily compare the performance of the membrane to those reported in the literature.

Thank you for the thoughtful suggestion; however, the thickness of the selective layers of an asymmetric hollow fiber membrane cannot be precisely defined, which may lead to misinterpretation of the membrane performance. Therefore, we decided to use GPU instead of Barrer to facilitate a more accurate and practical comparison between the membranes. The permeance (in GPU) and selectivity of the fabricated membranes were compared to those reported in the literature, as shown in Supplementary Fig. 1.

2. For the RWGS reaction, add a figure to plot CO₂ conversion as a function of temperature (H₂O permeance of the membrane) and weight hourly space velocity (WHSV) of the membrane reactors.

In accordance with your suggestion, Fig. 2 and its caption have been revised as follows:

Fig. 2

Fig. 2. Equilibrium-shift effect in the RWGS reaction. **a** CO₂ conversion as a function of temperature with (w/) and without (w/o) the membrane. **b** CO₂ conversion and amount of H₂O measured at the outlet of the sweep and reaction parts. **c** CO₂ conversion and amount of H₂O permeated from the reaction part to sweep part as a function of time on stream. **d** Additional amount of CO produced beyond equilibrium (Δ QCO) as a function of the amount of H₂O permeated from the reaction part to sweep part. Reactions were carried out using 6.4 g CuZn/Al₂O₃ catalyst under atmospheric pressure. Reaction and sweep gases flowed at 24 sccm with an identical composition of H₂/CO₂ ratio of 1 (gas hourly space velocity=225 mL/g_{cat} h). Fig. 2. Equilibrium-shift effect in the RWGS reaction.

3. Fig S.13: It does not make sense that when the sweep gas is changed to N₂, the CO₂ conversion drops significant below thermodynamic limits. As shown Table S2, the H₂O perm-selectivity of the membrane with respect to H₂ and CO₂ is close to 400, and 1400, respectively. These perm-selectivities do not seem to result in escaping of the reactants (H₂ and CO₂) to the sweep to cause lower CO₂ conversion. Authors need to clearly justify the experimental results of N₂ sweep. It might not be practical to use the sweep gas with the same composition of the feed, especially when the sweep gas involves the expensive H₂ in the RWGS reaction.

Please refer to our response to Comment 3 of Reviewer #1, which is closely related to your concern.

4. What is the effect of counter-current sweep to feed configuration on the CO₂ conversion of the RWGS membrane?

As you have mentioned, the flow configuration can significantly affect the separation-performance metrics such as permeance and selectivity in hollow fiber membranes because the ratio of the components and fluxes vary with the length of the hollow fibers (Journal of Membrane Science 125 (1997) 275-291, International Journal of Hydrogen Energy 31 (2006) 2243-2249). In this study, we have focused on demonstrating a high-temperature dehydration system using heat-resistible polymeric membranes in various chemical reactions. In-depth studies are being conducted on the membrane area, flow configuration, flow rate, and recirculation process with a view to optimizing and upscaling the membrane reactor and process. This content has been added to the relevant text, as follows:

(In Conclusions)

Future studies can be conducted on investigating the effects of the membrane area, flow configuration, flow rate, and recirculation process on the separation-performance metrics, with a view to optimizing and upscaling the membrane reactor and process.

5. Since membrane processes are slow, they need larger volumes to produce products. Authors need to compare the volume of a RWGS membrane reactor to that of a conventional packed-bed RWGS reactor to produce 1000 kmol/hr of syngas to be able to judge the practicality of membrane reactors at large scales in terms of size (umber of tubes, tube diameter, etc) and cost.

We agree that most low-temperature membrane processes (<100 °C) such as CO₂ capture, hydrogen purification, and air separation are slow. However, the membrane reactor used in this study focuses on dehydration for high-temperature reactions (>250 °C), which is advantageous for the rapid mass transfer of gas molecules through the membranes owing to the high diffusivity coefficient. The reactor size is determined by GHSV and the TR-PBO membrane used in the present study can permeate H₂O up to 0.56 mL/min/cm², which corresponds to the GHSV of 3024 mL/g_{cat}/h with an assumption of 27% conversion and 46% water permeation at the reaction temperature of 400 °C. Although the typical GHSV value of RWGS in a lab-scale reactor is about 6000 mL/g_{cat}/h (Juneau et al., Energy & Environmental Science 2020, 13, 2524), that in a commercial-scale reactor is expected to be lowered because of mass-transfer limitation of pelletized catalysts. Although the membrane reactor design and their operation should be optimized for

an up-scaled process, number of reactor tube was roughly calculated for the 1000 kmol/h of CO production. and tabulated on Supplementary Table 2.

(In the section “*Shifting the thermodynamic equilibrium*”)

In **Supplementary Table 2**, reactor volume of the membrane reactor is compared with that of packed-bed reactor for a commercial scale production of CO. The membrane reactor required less volume than the packed-bed reactor at the same feed flow rate.

(In Supplementary Information)

Supplementary Table 2. Comparison of the volume of TR-PBO integrated and packed-bed reactors for 1000 kmol/h production of CO. RWGS reaction temperature of 400 °C and the effective membrane area of $2.28 \times 10^5 \text{ m}^2$ are assumed based on the lab-scale reaction result. Reactor tube dimension: height = 11 m; diameter 4 inch. The water permeation rate at the reaction temperature of 400 °C allows the maximum feed flow of 3024 mL/g_{cat}/h for the TR-PBO membrane reactor, while feed flow of 3000 and 6000 mL/g_{cat}/h are applied to the conventional packed-bed reactor.

TR-PBO membrane reactor specification				
Water permeation rate @ 400 °C			mL/min/cm ²	0.56
Water removal ratio			%	46
Maximum feed flow			mL/g _{cat} /h	3024
Effective membrane volume per catalyst weight			cm ³ /g	0.0689
Specific volume of catalyst			cm ³ /g	0.79
Comparison				
		Membrane reactor	Packed-bed reactor	
CO production rate	kmol/h	1000	1000	1000
GHSV	mL/g _{cat} /h	3000	3000	6000
CO ₂ conversion	%	27	15	15
Catalyst weight	kg	55308	99555	49777
Catalyst loading per tube	kg	100.5	109.3	109.3
Number of reactor tubes		551	911	455

Minor comments:

1. Add a nomenclature to the manuscript.

Following your suggestion, we have added abbreviations in the revised manuscript as follows:

Abbreviations

6FDA, 4,4'-hexafluoroisopropylidene diphthalic anhydride; ATR-IR, attenuated total reflectance infrared; bisAPAF, 2,2-bis(3-amino-4-hydroxyphenyl)hexafluoropropane; DSC, differential scanning calorimetry; FE-SEM, field-emission scanning electron microscope; FID, flame ionization detector; FTO, Fischer–Tropsch Olefin Synthesis; GC, gas chromatography; Ha, absolute humidity; HAB, 3,3'-dihydroxy-4,4'-diamino-biphenyl; HPI, hydroxyl polyimide; LTA, Linde Type A; MFM, mass flowmeter; NMP, N-methyl-2-pyrrolidone; PAAC, poly(amic acid); RH, relative humidity; RWGS, reverse water–gas shift; SOD, sodalite; TCD, thermal conductivity detector; Td, thermal degradation, TGA, thermogravimetric analysis; Tg, glass transition temperature; THF, tetrahydrofuran; TR-PBO, thermally rearranged polybenzoxazole; XPS, X-ray photoelectron spectroscopy; XRD, X-ray diffraction; ZSM, Zeolite Socony Mobil.

2. Fig S1: Instead of references in brackets, show the actual name of the membranes on the figures.

Thank you for the suggestion. Owing to the space constraints in the figure, we have retained the reference numbers in brackets and have specified the names of the membranes in the figure caption.

Supplementary Fig. 1. (A) Water vapor permeance and (B) H₂/H₂O selectivity of inorganic membranes at 250–400 °C (7, 23-35). The numbers in the panels indicate supplementary reference numbers: [7, NaA zeolite membrane], [23, MFI zeolite membrane], [24, NAZSM-5 membrane], [25, ZSM-5 membrane], [26, Zeolite membrane], [27, mordenite membrane],[28, Zeolite 4A membrane], [29, NaA zeolite membrane], [30, A-type zeolite membrane], [31, Zeolite A membrane], [32, ZSM-5 membrane], [33, FAU-type zeolite membrane], [34, SOD and LTA membrane], and [35, SOD membrane].

3. Fig. 1: The numbers of the y-axis on the right-hand side of Fig. 1E should be added for the H₂O-permselectivity. There is no number on the right hand side y-axis.

We have revised Fig. 1e in accordance with your comment (please refer to our response to Comment 2 raised by Reviewer #1).

4. Some figures are not labeled properly. For example, in Fig. 2 C, two graphs are pointed to the left y-axis. Which set of data is for CO₂ conversion and which set of data is for H₂O permeance?

This point has been addressed by revising Fig. 2, as mentioned in our response to your major comment 2.

REVIEWER COMMENTS

Reviewer #1 (Remarks to the Author):

The authors have addressed the comments made to the former manuscript and have answered satisfactorily

Reviewer #2 (Remarks to the Author):

The authors appear to have addressed my comments (Reviewer #2) in their response. However, looking at the changes to the manuscript they introduce some new text that is problematic. I do not think that their membrane is suitable for water removal from the RWGS reaction.

'When pure N₂ was supplied as the sweep gas, the CO₂ conversion was lower than that obtained without the membrane because the N₂ sweep accelerated the partial permeation of the small reactant molecules through the membrane, before reacting on the catalyst bed. By using a sweep gas with the same composition as the reactants, the difference between the reactant concentrations on both the sides of the membrane could be reduced, thereby preventing the reactants from escaping and selectively removing the generated H₂O molecules. In practical applications, a simple moisture trap can be used to recycle the sweep gas. In Supplementary Table 2, reactor volume of the membrane reactor is compared with that of packed-bed reactor for a commercial scale production of CO. The membrane reactor required less volume than the packed-bed reactor at the same feed flow rate.'

Are small reactant molecules defined for this example? The use of a sweep gas with reactant molecules present suggests a significant problem with water permselectivity. This further suggests that this membrane would not be appropriate for this reaction. Simple recycling of the sweep gas would not be as straight forward as suggested; there would be a significant cost associated with this (and in the extreme case if you have a selective water adsorbent why not just use it on the reaction side and eliminate the need for a membrane completely?).

SI Fig 16(c), there appears to be some odd behaviour here? The material balances do not always close very well (being close to 90% at times) and one wonders if there are some other poorly understood processes occurring?

Reviewer #3 (Remarks to the Author):

No further comments.

Thank you for this opportunity to further improve our manuscript; we have addressed the reviewers' comments in this letter and in the attached document. The reviewers' comments (verbatim) and our point-by-point responses have been indicated in blue and black, respectively. The revised text has been indicated in red.

Reviewer #1 (Remarks to the Author):

The authors have addressed the comments made to the former manuscript and have answered satisfactorily.

Thank you for your review and comments.

Reviewer #2 (Remarks to the Author):

The authors appear to have addressed my comments (Reviewer #2) in their response. However, looking at the changes to the manuscript they introduce some new text that is problematic. I do not think that their membrane is suitable for water removal from the RWGS reaction.

'When pure N₂ was supplied as the sweep gas, the CO₂ conversion was lower than that obtained without the membrane because the N₂ sweep accelerated the partial permeation of the small reactant molecules through the membrane, before reacting on the catalyst bed. By using a sweep gas with the same composition as the reactants, the difference between the reactant concentrations on both the sides of the membrane could be reduced, thereby preventing the reactants from escaping and selectively removing the generated H₂O molecules. In practical applications, a simple moisture trap can be used to recycle the sweep gas. In Supplementary Table 2, reactor volume of the membrane reactor is compared with that of packed-bed reactor for a commercial scale production of CO. The membrane reactor required less volume than the packed-bed reactor at the same feed flow rate.'

Are small reactant molecules defined for this example? The use of a sweep gas with reactant molecules present suggests a significant problem with water permselectivity. This further suggests that this membrane would not be appropriate for this reaction. Simple recycling of the sweep gas would not be as straight forward as suggested; there would be a significant cost associated with this (and in the extreme case if you have a selective water adsorbent why not just use it on the reaction side and eliminate the need for a membrane completely?).

We acknowledge the reviewer's comment that the reported membranes do not exhibit perfect selectivity in current reactor systems. However, we believe that this is not a problem with the membrane itself but a challenge from a reactor design perspective. Specifically, when N₂ is used as the sweep gas, a section where CO₂ and H₂ partially crossover is observed in the upper part of the reactor, before the reaction gas meets the catalyst. This is because the permeability of CO₂ or H₂ decreases by an order of magnitude in the section where the reaction takes place in the catalyst layer, while the selectivity of H₂O/H₂ and H₂O/CO₂ for the mixed gas is maintained (Supplementary Fig. 12). Supplementary Figure 13 shows that the reactor used in this study has a small volume of the catalyst layer, making this phenomenon more conspicuous. However, in large-scale reactors, where the catalyst fills most of the space with few empty spaces, the issue of reactant crossover is expected to be minimized. Nevertheless, if a reactant is to be used as a sweep gas, it is still more practical to use a simple gas-liquid separator rather than an adsorbent, as the former can continuously separate water from gas by chilling the chamber, while the latter requires periodic shutdown of the process for high-temperature regeneration.

The paragraph indicated by the reviewer has been revised without using the "small reactant molecules" term as below:

“When pure N₂ was used as the sweep gas, the CO₂ conversion was lower than that obtained without the membrane. However, we believe that this is not a problem with the membrane itself, but rather a challenge from a reactor design perspective. Supplementary Fig. 12 shows that in the presence of H₂O, the permeabilities of CO₂ and H₂ decrease by an order of magnitude compared to their pure-gas permeabilities, while H₂O/H₂ and H₂O/CO₂ permselectivities are maintained. This suggests that CO₂ and H₂ crossover occurs in the upper part of the reactor before the reaction gas meets the catalyst. This phenomenon is likely more apparent in the lab-scale reactor used in this study due to the small volume of the catalyst bed, as shown in Supplementary Fig. 13. However, this issue is expected to be minimized in large-scale reactors where the catalyst fills most of the space with few empty spaces. Moreover, this issue can be mitigated by using a sweep gas with the same composition as the reactants, reducing the difference in reactant concentrations on both sides of the membrane and preventing the reactants from escaping while selectively removing generated H₂O molecules. In this case, a simple gas-liquid separator can be used in practical applications to continuously separate water from the gas and recycle the sweep gas. In Supplementary Table 2, reactor volume of the membrane reactor is compared with that of packed-bed reactor for a commercial scale production of CO. The membrane reactor required less volume than the packed-bed reactor at the same feed flow rate.”

SI Fig 16(c), there appears to be some odd behaviour here? The material balances do not always close very well (being close to 90% at times) and one wonders if there are some other poorly understood processes occurring?

Due to the low concentration of methane (0.4%) used in the methane oxidation reaction, there were experimental errors in measuring the absolute amount of unreacted methane and carbon dioxide generated after the reaction. Specifically, when using a membrane for methane oxidation, a multiposition valve was installed at the rear end of the reactor to alternately measure the two outlet gases of the reaction part and the sweep part, which led to a slight measurement error during operation. We added this comment to Note 5 in the Supplementary Information as below:

“We note that since the concentration of methane used in the methane oxidation reaction is low (0.4%), there was a slight experimental error in measuring the absolute amount of unreacted methane and carbon dioxide generated after the reaction. A multiposition valve was installed at the rear end of the reactor when using a membrane for methane oxidation to alternately measure the two outlet gases of the reaction part and the sweep part, which resulted in a slight measurement error during operation.”

Reviewer #3 (Remarks to the Author):

No further comments.

Thank you for your review and comments.

REVIEWERS' COMMENTS

Reviewer #2 (Remarks to the Author):

Of course the authors are correct that you can engineer the membrane reactor to mitigate the problems with membrane selectivity. However, this arrangement is now quite complex and I would not be surprised to see some unanticipated behaviour if further more detailed work were undertaken (which is really what should happen now). The example is no longer the simple example that the authors first intended I suspect. It would now be better as the subject of an original research article covering only this mode of operation.

I do not understand the statement, 'However, this issue is expected to be minimized in large-scale reactors where the catalyst fills most of the space with few empty spaces.'

Reviewer #2 (Remarks to the Author):

Of course the authors are correct that you can engineer the membrane reactor to mitigate the problems with membrane selectivity. However, this arrangement is now quite complex and I would not be surprised to see some unanticipated behaviour if further more detailed work were undertaken (which is really what should happen now). The example is no longer the simple example that the authors first intended I suspect. It would now be better as the subject of an original research article covering only this mode of operation.

I do not understand the statement, 'However, this issue is expected to be minimized in large-scale reactors where the catalyst fills most of the space with few empty spaces.'

The manuscript currently highlights the promise of membrane reactors for a range of water-producing reactions. However, as per the reviewer's recommendations, we are directing our attention towards enhancing the design of the reactor and improving membrane efficiency, both of which are crucial factors for facilitating its widespread industrial application.

The sentence indicated by the reviewer has been revised as below:

“However, this issue is expected to be minimized in large-scale reactors, where the catalyst bed occupies the majority of the reactor volume, leaving only a small dead space above the catalyst bed where only membranes are present.”